# Physical limits of sea-level rise adaptation in global river deltas

Kiara G. Lasch [1] ✉, Jaap H. Nienhuis [1], Gundula Winter [2] & Marjolijn Haasnoot [1,2]

Sea-level rise threatens deltas worldwide, requiring adaptation to flood risks. Delta adaptation is typically presented as a choice between five strategies: advance, protect-closed, protect-open, accommodate, and retreat. However, a full assessment of the physical feasibility of these strategies across deltas remains limited. We present a first-order assessment of the physical solution space for adaptation to sea-level rise for nearly 800 deltas globally. Here, we show that current technologies, resources, and space provide at least one physically feasible delta-wide strategy for every delta to adapt by 2100. This number may increase in the future through technical innovation or collaboration between deltas. The type and number of physically feasible strategies are mostly determined by delta's physical characteristics, whereby large, urbanized, or frequently flooded deltas have fewer options than small, rural, or infrequently flooded deltas. Our analysis highlights the risk of resource limitations as global deltas will need to adapt simultaneously to future flood risks.

Deltaic inhabitants are vulnerable to floods, droughts and land loss due to climate change[1,2], with sea-level rise (SLR) expected to pose the largest challenge to these coastal areas[3]. Such impacts, combined with a rapidly expanding population, necessitate a long-term commitment to adaptation[2,4,5]. Several studies have assessed adaptation at the delta scale[6–19]. For example, in the Netherlands, a feasibility assessment showed that the construction lifetimes of barriers and pumping stations will pose significant challenges[17,20] and in the United States, an analysis of location, damages, hazards, and socioeconomic and demographic characteristics informed voluntary buyouts of flood-prone properties[18].

Global assessments of coastal adaptation are rare, but can aid local-scale assessments. To date, global studies include evaluations of flood protection costs and benefits[21], timing of adaptation needs[13,22] and currently implemented adaptation strategies[23]. The feasibility of adaptation has also been assessed across different dimensions, by identifying barriers to adaptation options in different geographical and socio-economic settings[24]. Yet, a physical feasibility assessment of adaptation strategies for global deltas under future flood risks remains limited.

In this context, mapping the physical solution space of deltas could inform the physical opportunities and constraints for delta adaptation at a global scale. The solution space is the 'space within which opportunities and constraints determine why, how, when and who adapts to climate risks'[25–27]. In addition to the physical adaptation potential, resource limitations may also be triggered, as all deltas are vulnerable to the risks of SLR and may require synchronous adaptation in the future[28–30]. Resource coordination and innovation can therefore be guided and targeted through a global assessment of adaptation.

We categorise delta adaptation into five flood risk adaptation strategies. Following the IPCC framework, we distinguish protect, accommodate, advance and retreat[2,31]. We split the protection strategy into two, as, for deltas, a fundamental decision involves whether the river remains open to or closed off from the sea[12,17], resulting in the protect-open and protect-closed strategies, respectively (Fig. 1; see Supplementary Text 1 for an extended description of strategies and examples in practice). We introduce a method to map the physical solution space (PSS) of nearly 800 deltas to adapt to SLR. The PSS is mapped under three SLR scenarios following the IPCC's SSP-RCP scenarios[32] and across the aforementioned adaptation strategies,

[1]Department of Physical Geography, Faculty of Geosciences, Utrecht University, Utrecht, the Netherlands. [2]Deltares, Delft, the Netherlands.
✉ e-mail: k.g.lasch@uu.nl

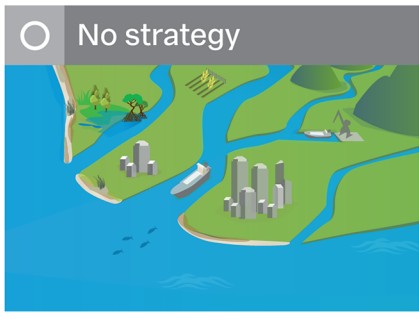
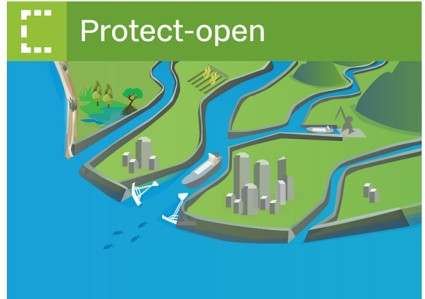
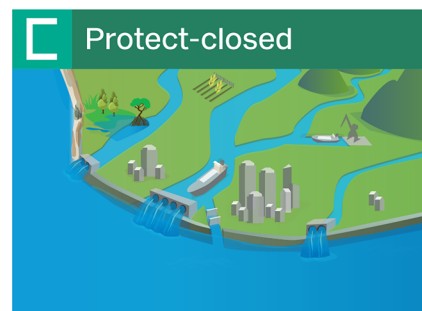
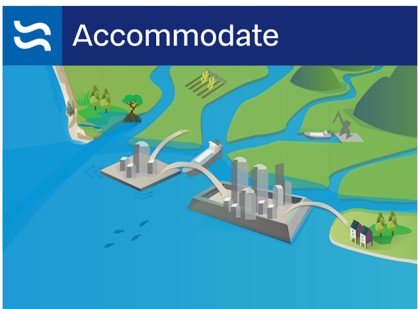
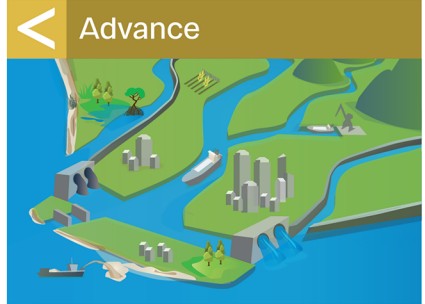
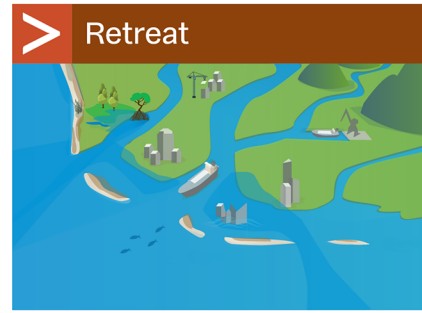

**Fig. 1 | Six categories of delta adaptation strategies to reduce flood risk and adapt to sea-level rise, defined based on the IPCC's four strategies, namely, protect, accommodate, advance and retreat[2,31].** No strategy represents deltas without implemented flood risk adaptation measures. A protect-open strategy protects inland areas from flooding, but maintains an open connection to the sea by constructing engineered structures, such as levees, along the coast and rivers and (potentially) building storm surge barriers at the river mouths which close during storm surge events. Protect-closed aims to protect inland areas by constructing engineered structures, such a levees, along the coastline, while closing off the connection with the sea, installing pumps to pump water from the river to the sea and using retention areas to store excess river discharge during high river flows. Accommodate adopts a 'living with water' concept, whereby at-risk areas are adjusted to reduce the vulnerability to flooding (e.g. through elevating buildings or constructing floating houses). Advance involves the extension of the coastline seaward beyond a coastal lagoon to build flood defences and create new land. Retreat focuses on a planned and permanent relocation of people and assets to reduce exposure to coastal hazards. Supplementary Text 1 provides examples in practice.

assuming the strategy is adopted across the entire delta (see 'Methods' and Supplementary Text 2 for a description of the delta areas). We map the PSS for the year 2100 to allow sufficient time for planning and implementation, given that SLR adaptation strategies typically have long lifespans and lasting impacts on society[33]. For each adaptation strategy, we create physical indicators of feasibility, which represent quantifiable requirements of key measures, namely: material requirements and availability for creating a new coastline (advance) or building coastal and/or riverine levees (protect), the river mouth width for constructing a storm surge barrier (protect-open), the river pump capacity and excess water storage capacity of retention areas (advance and protect-closed), the flood depth for elevating infrastructure and buildings (accommodate) and the available space for relocating urban areas (retreat) (see 'Methods'). We assess the physical feasibility of adaptation strategies by applying three thresholds to each physical indicator (see 'Methods'). If no thresholds are exceeded (i.e. there are no technological or physical limits), a strategy could be available to implement to reduce future flood risks. The thresholds chosen represent three scales of adaptation: (1) the current known measures, reflecting the largest or commonly used scales of measures in practice, (2) a low-resource (material) version of the current known measures and (3) an innovative version of the current known measures reflecting larger, more resource-demanding measures that would require technological advancements or collaboration to adopt (see Supplementary Dataset 1 for a database of existing examples of adaptation measures and supplementary Text 4 for rationale to support the thresholds). The low-resource measures are only physically feasible under conditions of restricted resources, which may limit the range of adaptation strategies available for implementation in deltas. In contrast, the innovative thresholds expand opportunities for adaptation and increase the range of physically feasible adaptation strategies, but only if technological advancements occur or if sufficient resources are available. Ultimately, whether a delta meets these thresholds determines the number and type of physically feasible adaptation options under future SLR.

In this work, we assess adaptation to SLR to reduce coastal flood risks while accounting for river flooding. We provide a first-order assessment of the PSS for nearly 800 deltas, showing that for every delta, at least one delta-wide adaptation strategy is physically feasible to implement following three SLR scenarios. However, the number and type of physically feasible adaptation strategies often differ across deltas based on the deltas' physical characteristics and the scale of adaptation. By mapping the PSS of adaptation strategies, we provide insight into the technical innovation or coordination that may be necessary to address the impacts of future SLR in deltas.

## Results
### A global assessment of delta adaptation solution space
Following three SLR scenarios[32] that span a wide range of flood-risk futures for 2100, we find that for every delta, physically feasible options exist with current space, technologies or resources, with at least one adaptation strategy available for SLR adaptation. Under an intermediate SSP2-4.5 global warming scenario specifically (median estimate of 0.47 m SLR globally in 2100[34,35]), we find that even under limited technical or resource conditions, at least one adaptation strategy is physically feasible for nearly every delta (except for the Rhine-Meuse; see Results for further elaboration), with all five strategies physically feasible for approximately 28% of deltas in their most low-resource form. Conversely, though, under conditions of increased technical

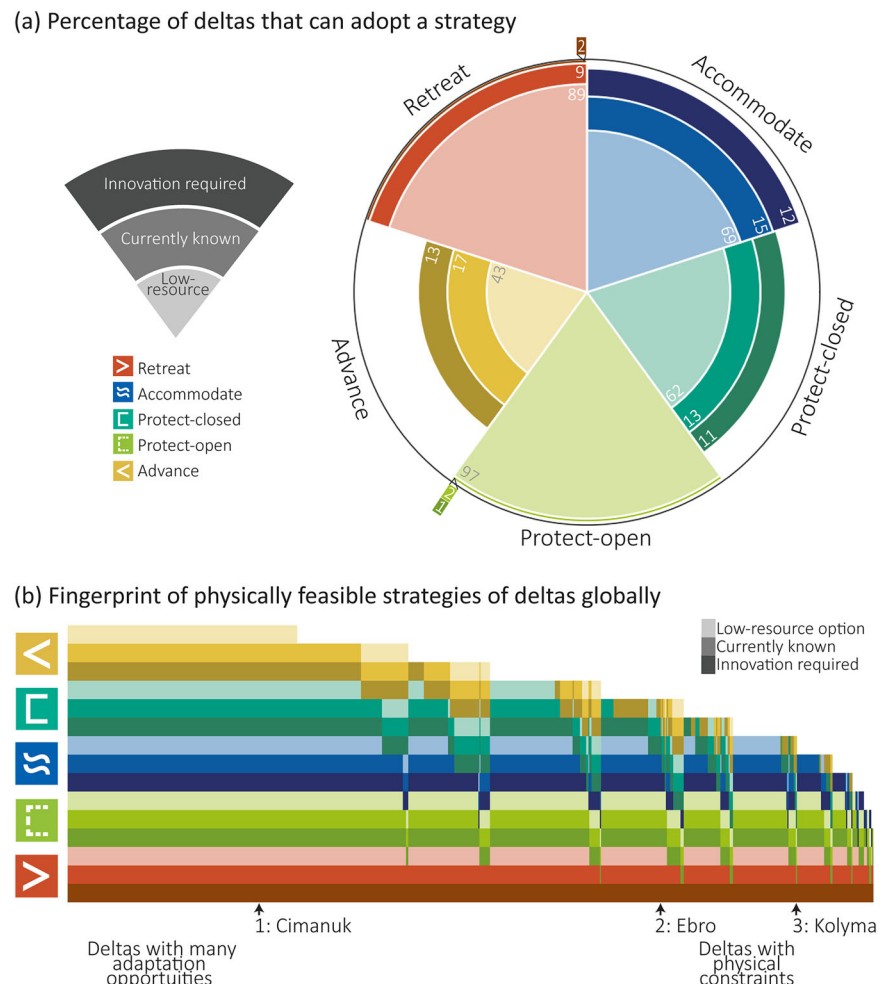

**Fig. 2 | A global analysis of the types of physically feasible adaptation strategies for 769 deltas. a** The percentage of deltas that can adopt each adaptation strategy under the three physical indicator thresholds following an SSP2-4.5 scenario. The outline of the circle implies 100% of deltas (769 deltas). The white space in the radar plot represents deltas that cannot adopt a certain strategy. **b** The fingerprint of delta adaptation showing the physically feasible strategies per delta, across three physical indicator thresholds. Each column represents an individual delta, with the left columns showing deltas where all strategies are physically feasible across all thresholds, while deltas towards the right are more physically constrained in the strategies they can adopt.

innovation, the number of physically feasible strategies expands to at least two per delta, with approximately 68% of deltas having all five adaptation strategies physically feasible to implement.

The global fingerprint of delta adaptation shows that the number and type of physically feasible adaptation strategies varies between deltas and across all scales of adaptation potential (i.e. across all three thresholds; Fig. 2). Following an SSP2-4.5 scenario by 2100, the physical indicators reveal that several strategies are commonly physically feasible across deltas, with protect-open (99% of deltas), accommodate (84% of deltas) and retreat (98% of deltas) being physically feasible with currently known resource and technological capabilities and available space, requiring no innovation for delta-wide implementation (Fig. 2a). The remaining deltas face physical constraints that limit the implementation of these strategies (protect-open: 1%, accommodate: 12%, retreat: 2%) and require technological innovation or collaboration to make them physically feasible to implement. Innovation for these strategies includes storm surge barriers that exceed 9 km in length, home raising of more than 1 m, or planned relocations outside of the delta area due to a lack of space, respectively. We find that protect-closed and advance strategies face greater physical limits based on our indicators (Fig. 2b). While current pumping capacities are sufficient to manage mean river discharge for the majority of global deltas (91%), these capacities are exceeded

during high river flows, necessitating larger pumping capacities or the storage of excess discharge in designated retention areas. Under a 99th percentile river flow event[36,37], protect-closed is physically feasible for most deltas (75%) as the discharge can either be entirely pumped out based on current pump capacity capabilities, or excess discharge can be stored in retention areas. Alternatively, excess river water can be drained under gravity. However, future SLR may limit gravity drainage, which may increase pump capacity requirements beyond what is currently possible and further constrain the physical feasibility of protecting these deltas. Notably, advance is the least physically feasible strategy globally, with 27% of deltas lacking the necessary physical conditions for its implementation based on the indicators, even under assumed technical innovations (Fig. 2a). Moreover, despite having greater water retention capacities from coastal lagoons than protect-closed, advance is physically constrained by material limitations to aggrade offshore coasts, such as insufficient river sediment or offshore depths exceeding 30 m (Fig. 2a). According to our indicators, advance is only physically feasible when both the pump capacity (based on the annual mean river discharge and the 99th percentile river discharge with sufficient retention areas) and the material availability (based on efficient river sediment collection or accessible offshore depth) thresholds are met (see 'Methods'). However, a small river discharge and a large sediment discharge are not

concurrent in deltas[38] and deeper offshore depths physically constrain sand collection for coastline extensions.

## Physical characteristics shape adaptation feasibility

Delta characteristics such as surface area, land use and flood extent shape the adaptation potential, leading to clustering in the number and type of physically feasible strategies across deltas. Generally, small deltas (<100 km² surface area) with low future flood extents (<50% of their surface area) have large PSSs, including low-resource measures

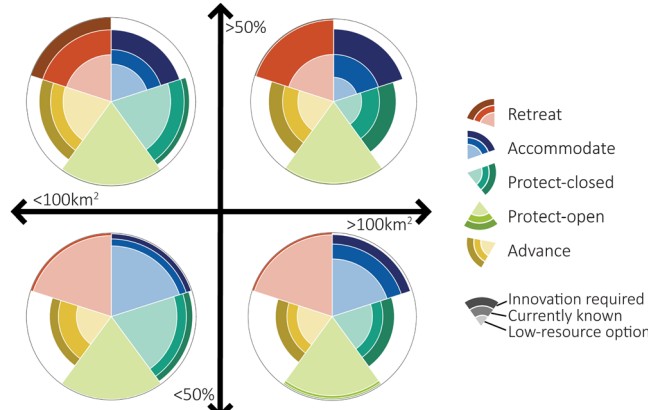

**Fig. 3 | The percentage of deltas that meet the strategy-specific thresholds across various combinations of physical characteristics following an SSP2-4.5 scenario.** Physical characteristics are geomorphic area of the whole delta (less than 100 km² or greater than 100 km²) and flood extent (less than 50% or greater than 50%) of the total geomorphic area. The outline of each circle implies 100% of deltas and the white space within the radar plot represents deltas that do not meet any of the strategy-specific thresholds.

(Fig. 3), as seen in the Buller delta (New Zealand) and the Mangoro delta (Madagascar)(Fig. 4). These deltas are characterised by narrow river widths, which make low-resource (narrow) storm surge barriers (protect-open) physically feasible to implement. Similarly, low coastal flood extents imply there is non-flooded land available, making a less intensive retreat from flooded areas physically possible. However, if the flood extent (and thus also flood depth) in small deltas increases (>50% of its surface area), retreat and accommodate strategies become physically constrained (Fig. 3). In such cases, a planned retreat outside of built-up areas or elevating urban infrastructure by more than 1 m on stilts may be required, as in the Pineios delta (Greece). In the Jiulong delta (China), retreating outside of the delta area and elevating infrastructure by more than 2 m may be necessary under future flood risks (Fig. 4).

The adaptation potential of larger deltas (>100 km²) with low flood extents (<50%) is often physically constrained unless substantial technical advancements are developed (Fig. 3). Examples include the Mississippi, Niger, Indus and the Ganges-Brahmaputra-Meghna deltas (Fig. 4), where technological innovation is required for the implementation of several strategies given the large physical characteristics of these deltas, which scale with the size of the delta[39]. In larger deltas, large mean river discharges exceed both current and innovative pump capacities and the volume of water that needs to be stored during high river flows exceeds the storage capacity despite larger areas, thereby limiting the physical feasibility of protect-closed and advance strategies, even with innovation. This may require an alternative strategy or a more targeted approach, such as concentrating technological advances in at-risk areas or developing innovative measures at a smaller scale. However, the low flood depth and flooded areas in these deltas make accommodating and retreat strategies more physically feasible. In the Mississippi delta, for example, small-scale community relocations have already taken place[40,41] and our analysis supports that a low-resource retreat is physically feasible for this delta. Additionally, the

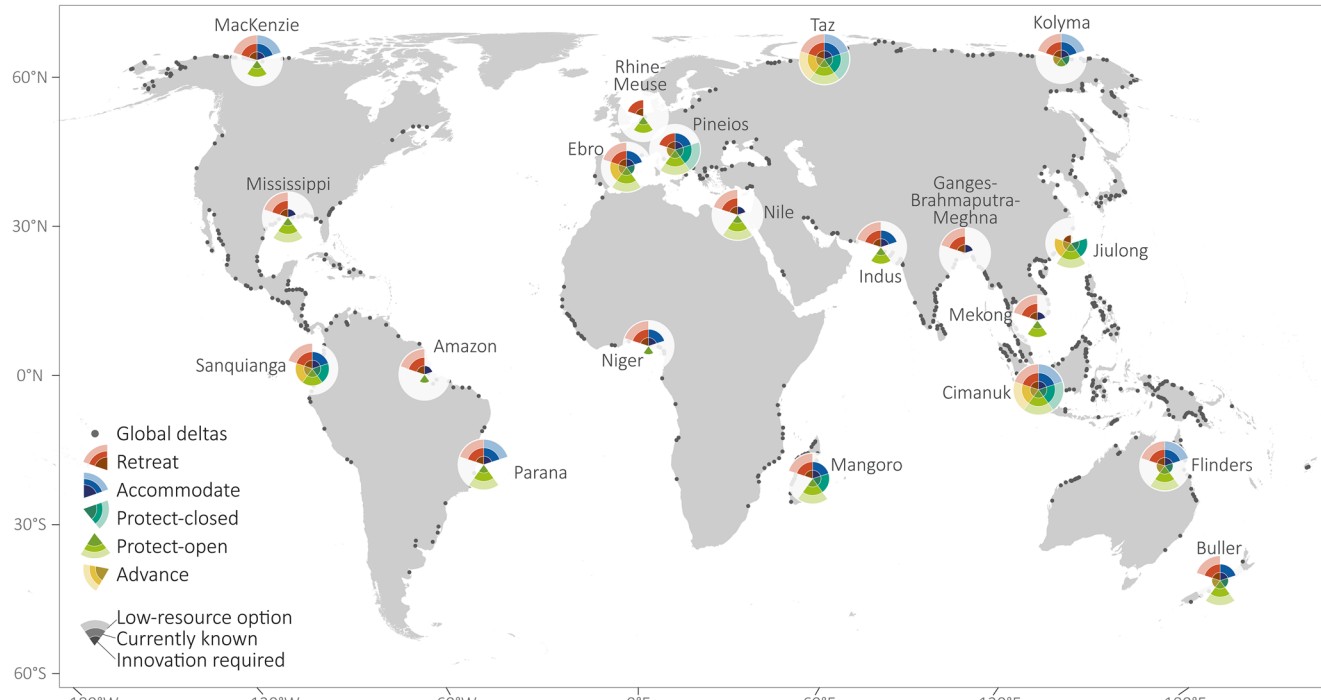

**Fig. 4 | A selection of 20 delta's physical solution spaces based on three physical indicator thresholds under an SSP2-4.5 scenario.** The outer bar represents strategies that are physically feasible under limited resource conditions, the middle bar represents strategies that are physically feasible using current scales of implemented measures and the inner bar represents strategies that are only feasible with innovation or technological advancements. Grey dots represent the remaining 749 deltas whose physical solution spaces were also mapped, but are not visualised here. The blank basemap was created using publicly available World Countries shapefile data in QGIS (10m Cultural Vectors | Natural Earth Data)[98].

construction of levees and the Inner Harbour Navigation Canal (IHNC) Surge Barrier protecting the city from high water levels in Lake Borgne (Mississippi)[42] aligns with our findings that the protect-open strategy is physically feasible based on the river width of this delta.

If the flood extent of large deltas increases (>50%), accommodate and retreat become less physically feasible, given the challenges associated with accommodating areas with greater flood depths and relocating urbanised areas to limited non-flooded areas, respectively (Fig. 3), as seen in the Rhine-Meuse delta (the Netherlands) (Fig. 4). The PSS of the Rhine-Meuse delta is one of the smallest in our global assessment, whereby, based on our physical indicators, none of the low-resource measures are physically feasible to be adopted delta-wide. To date, some of the largest scales of measures have been implemented in this delta and many local-scale assessments of adaptation in the Netherlands indicate that there are adaptation opportunities[17,43]. This illustrates how the delta's scale influences adaptation opportunities and constraints. The Rhine-Meuse delta's large mean and peak river discharge, river width, flood depth and flood area relative to its total area and to other deltas, limit the adaptation potential based on the indicators in this assessment, despite recent studies indicating that these strategies are technically possible with substantial innovation[44,45]. While our thresholds are typically based on the largest existing implemented scales of measures, assessing the physical feasibility of delta-wide strategies may risk overlooking the adaptation potential (see 'Discussion').

We also find that land-use shapes the adaptation potential in deltas. Deltas with large urban areas generally have contracted PSSs, necessitating technical innovation to adapt to SLR (Fig. 5). This is especially evident in the accommodation strategy, where physical feasibility is based on the height required to elevate urban flooded areas. For example, in highly urbanised deltas, such as the Mississippi, the Ganges-Brahmaputra-Meghna and the Rhine-Meuse deltas (Fig. 4), innovation to elevate infrastructure by more than 1 m and sometimes 2 m may be required. For these deltas, protective strategies may be more preferable if the physical characteristics are suitable. These findings align with other studies, which show that urbanised deltas may opt for more protective measures if there is sufficient space and resources to construct them[6,10,14,20].

In comparison, rural deltas with low population densities tend to have larger PSSs, supporting numerous physically feasible strategies in their lowest-resource forms (Fig. 5), as seen in the Flinders delta (Australia) and Taz delta (Russia)(Fig. 4). However, rural deltas may opt for a 'living with water' approach (accommodate) or a planned relocation to higher ground (retreat)[6] instead of protective measures[15].

## Global material requirements and potential conflicts and limits

The global pursuit of flood risk adaptation may be restricted by material needs. Following an advance strategy, we find that the material required to extend every delta's coastline 8 km seaward is estimated at ~3099km³, which greatly exceeds the available fluvial sediment supply across global rivers over 75 years (up to 2100). This points to the danger of resource limitations and potential competition for resources if this strategy becomes widespread among deltas[4].

Deltas may be able to meet their material requirements using their own resources, which is often observed in practice[28,46–48]. We find that many deltas (79%) are material-sufficient with currently existing nourishing technology, meaning they can advance their coastline using local material, either by collecting natural river sediment or from extracting offshore sand (Fig. 6). However, even if deltas can source sufficient material from their rivers or offshore coast, the pump capacity requirements may pose additional challenges. While effective river sediment collection may be sufficient to aggrade the new coastline, a large sediment discharge is typically simultaneous with a large

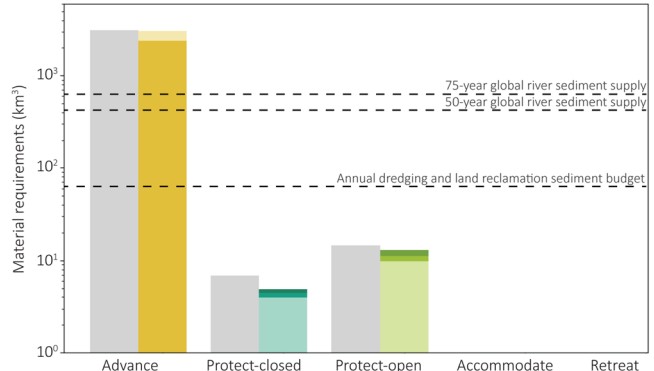

**Fig. 6 | Sum of the global material requirements (km³) for each adaptation strategy following an SSP2-4.5 scenario.** The grey bars indicate the total material requirements if every delta adopts the strategy. The dark yellow bar (advance) represents the material requirements for material-sufficient deltas. The lighter yellow bar (advance) represents the material requirements for material-deficient deltas that need to outsource material. The shaded bars for protect-closed and protect-open represent the material requirements for deltas under three strategy-specific thresholds. The grey dotted lines represent the 75-year and 50-year global river sediment supply and the annual global dredging and land reclamation sediment budget in 2010[38]. Note that these material requirements constitute a conservative estimate based on the volume of protection measures and do not take into account potential maintenance nourishment, for example.

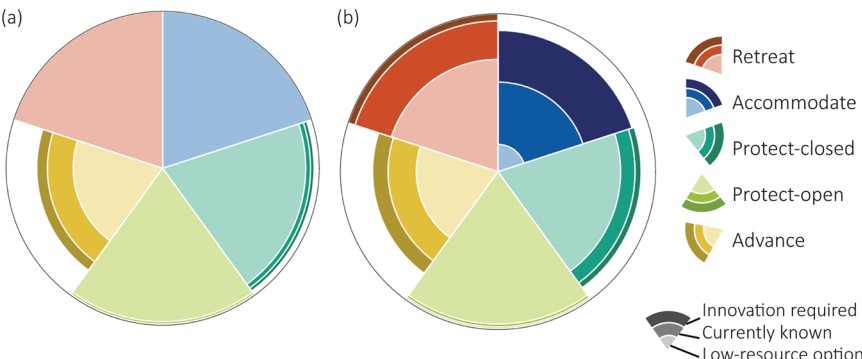

**Fig. 5 | The percentage of rural versus urban deltas that meet the strategy-specific thresholds following an SSP2-4.5 scenario.** The physically feasible adaptation strategies for **a** rural deltas with low population densities and **b** urban deltas with densely populated areas. The outline of each circle implies 100% of deltas and the white space within the radar plot represents deltas that cannot adopt a certain strategy.

river discharge, making pump capacity requirements unfeasible, thereby limiting the advance strategy's physical feasibility[38]. The reverse situation is also possible, where deltas may meet the pump capacity requirements but are material-deficient and cannot collect sufficient material to extend their coastline, either due to large offshore depths (>60 m) or low sediment retention rates (<20%).

Some deltas (8%) can only advance their coastline under innovative material collection conditions and the remaining deltas (17%) are material-deficient to pursue advance and will require import (Fig. 6). The total material required for material-deficient deltas amounts to approximately 2409km³, which is nearly 40 times larger than the annual global sediment budget from all dredging and land reclamation activities (9.8Gt per year, 61.2km³) in 2010[38] and far exceeds both the 50-year and the 75-year global river sediment supply (~416km³ and ~624km³, respectively)(Fig. 6). In these cases, alternative adaptation options or external materials, potentially sourced from other shallow seas, may be required. This highlights that adaptation efforts surpass local boundaries and collaboration between countries through knowledge and resource sharing may be necessary in the future[4].

Protect strategies have substantially lower material requirements compared to the advance strategy, increasing their physical feasibility. The levee material requirements are ~6.81km³ for protect-closed and over double this, ~14.56km³, for protect-open (Fig. 6). These material requirements are considerably lower than the total quantity of sediment naturally delivered from rivers. As such, a portion of the sediment supply could be allocated to levee construction if the pump capacity or storm surge barrier width thresholds are also met, respectively.

### Physical characteristics and adaptation thresholds versus climate change

We find that higher sea levels increase flood risks and shrink the PSS, in agreement with previous studies[13,31,49]. However, this change is relatively small compared to our modelled differences in physical feasibility based on the scale of adaptation measures (i.e. thresholds). We find that the scale of the adaptation measures has a greater influence on the physical feasibility of adaptation strategies than the SLR projections for 2100 (see Supplementary Text 5 and Supplementary Fig. 4 for a comparison between climate scenarios and thresholds). For instance, the scale of raising infrastructure following accommodate, by 0.5 m, 1 m or 2 m, exceeds the median projected global mean SLR for 2100 under SSP1-2.6, SSP2-4.5 and SSP58.5 scenarios (0.4 m, 0.47 m, 0.63 m, respectively)[35]. Moreover, the feasibility of offshore dredging to collect sand at depths of 15 m, 30 m, or 60 m is more constrained by technological capabilities to reach the sand than by a 0.5 m SLR.

Our assessment further indicates that the current physical characteristics of the deltas (e.g. size, elevation, river discharge as used in this study) are a stronger determinant of the physical feasibility of strategies by 2100 than the projected SLR. For example, the physical feasibility of flood risk protection measures, such as storm surge barriers, depends more on the river width, which varies from narrow (100 m) to wide (10,000 m) across deltas, than on SLR following future climate scenarios by 2100. Thus, in deltas with wide rivers, protect-open is more physically constrained by the delta characteristics than SLR. Building on this understanding, some physical characteristics within the delta have stronger influences on the type of physically feasible strategies than others. More specifically, the physical feasibility of the advance and protection strategies is largely dependent on physical indicators that consider the geomorphological characteristics of the delta, including coastline length, offshore depth, sediment discharge, potential water retention areas and river length, width and discharge. For example, the physical feasibility of protect-closed is based on the pump capacity and water retention area indicators, which are physically constrained in large deltas with large river discharges that cannot be pumped or stored in water retention areas under high

river flow events. For other strategies, such as accommodate and retreat, physical feasibility is more dependent on the land use and flood extent of the delta. For example, the physical feasibility of an accommodation strategy is determined by the flood depth. Ultimately, assessing the magnitude of the physical characteristics in deltas allows for a quick identification of strategies that may be physically suitable or constrained to implement.

Under higher amounts of SLR beyond the likely range for this century or after 2100, SLR is expected to have a larger influence on the PSS (see Supplementary Text 5 and Supplementary Fig. 5 for sensitivity analysis with higher SLR). Moreover, climate change is expected to alter future river and sediment discharge[50–52], which may impact pump capacity and water retention requirements (protect-closed and advance) and sediment availability (advance). Our sensitivity analysis reveals that changes in these parameters, based on literature[50–52], impact the PSS of several deltas, which influences local decision-making. Increased river discharge values typically constrain the PSS where protect-closed and advance become physically unfeasible without innovation, for example, in tropical (monsoon) regions[50]. Conversely, decreased discharge projections typically expand the PSS, making both advance and protect-closed physically feasible with current technological capabilities and resources, for example, in North America[50,51] (see 'Methods' and Supplementary Text 6 for sensitivity analysis results and examples).

## Discussion

We find that current technologies, resources and space unlock opportunities for adaptation in global deltas (Fig. 2), which is consistent with previous findings that viable options to adapt already exist[49]. However, some deltas have contracted PSSs compared to others. In particular, large, urbanized and frequently flooded deltas are constrained in their material availability and technological feasibility (Fig. 3). Such constraints may lead to competition for global resources[4,53], but also point to the importance of coordination and collaboration between deltas and their surrounding communities to expand their PSS. For instance, a material-sufficient delta, with substantial natural river sediment or accessible offshore sand, could allocate resources to areas with a greater need for those resources (e.g. Singapore)[53]. Alternatively, upstream dams could be removed to boost sediment delivery in delta catchment areas (e.g. Mekong delta)[54]. While these approaches may be costly, they provide greater adaptation opportunities and broaden the PSSs of constrained deltas.

Resource coordination in deltas also depends on the timing of adaptation needs, whereby understanding when the risks are the largest allows for more effective resource allocations and prioritisations. Since adaptation timing and commitment rates vary per country[4], assessing the timing of adaptation needs in deltas may require a further local-scale assessment of the PSS. Moreover, innovating and implementing large adaptation measures at a large scale will likely require long lead times[4,49]. If strategy implementation is left too late and only strategies with short lead times are available, the PSS may contract further. Thus, projecting the PSS into the future supports near-term decision-making given the different lead times of strategy implementation[4,55]. Ultimately, understanding these timelines in deltas is essential to allow sufficient lead times for developing technological and innovative solutions and advancing societal readiness.

To address uncertainties associated with climate change, an adaptive approach is recommended[33,56–59]. In deltas, low-resource measures (e.g. localised pumps or small-scale land reclamation) can initially be implemented if available, which maintains flexibility in decision-making. If conditions change, for example, if flood risks increase or if more resources become available, measures can be upscaled or upgraded. Moreover, having multiple options, especially when starting with low-resource measures, may provide opportunities to switch to alternative strategies if an adaptation tipping point is

reached, thereby reducing the possibility of a lock-in situation[17,26,33]. Such flexibility is beneficial for ensuring long-term adaptability and enabling timely responses to future flood risks.

To work towards a holistic solution space assessment for delta adaptation, all possible adaptation measures, potential hazards and different dimensions within the solution space need to be considered. We assess the physical feasibility of a strategy being implemented across the entire delta. However, at a local scale, maintaining and building upon existing measures may be more feasible, in terms of financial and material requirements, than adopting a new strategy. Alternatively, spatially targeted strategies that are tailored to the flood risks in the delta, or hybrid approaches that combine multiple strategies to address different challenges[12], will further expand the PSS. For example, in the Mississippi delta, both structural adaptation measures, such as levees and barriers and ecosystem-based approaches, like wetland restorations, are implemented alongside the elevation or relocation of homes to deal with coastal flooding[40–42,60,61]. In the Ganges-Brahmaputra-Meghna delta, levees to protect the inland areas[62] and controlled flooding to raise the land[63] are combined with cyclone shelters with early warning systems[64].

The PSS can also contract or expand when considering additional delta hazards beyond SLR-driven flood risks. For example, salinity intrusion may require land use changes, which can affect where and how structural defences are built. Alternatively, nourishment strategies to offset SLR-induced erosion may consume available sediment, which may deplete the supply available to accommodate or advance strategies. Finally, socio-economic developments, such as urbanisation, can change the PSS in terms of the space available for raising levees or space to retreat to.

Within the physical dimension of the adaptation solution space, a key component is understanding the limits of what is veritably physically feasible and identifying the technological advancements or collaborative capacities that can extend these boundaries, thereby expanding the PSS. However, beyond the physical dimension, the social, cultural, legal and governance dimensions play a large role in adaptation feasibility[65]. As such, physical feasibility does not necessarily imply social or political acceptance, or financial or environmental feasibility[5,12,25]. From a societal perspective, there may be different preferences for ecosystem services and nature preservation versus grey infrastructure. Some adaptation strategies may be physically feasible, but may have costs in terms of ecosystem services. Alternative nature-based solutions, which provide both flood risk reduction and a range of ecological benefits, may therefore be preferred[66]. Additional societal constraints, which include challenges associated with institutional fragmentation, social inequalities and low public awareness, may further limit strategy implementation and acceptance. Moreover, economic barriers, such as financial or social conflicts, may limit the adaptation potential before physical limits arise[43,62,67]. These trade-offs may shrink the solution space further. For example, while protect-open is physically feasible across the majority of deltas to protect against coastal and river flooding (Fig. 2), it is a costly strategy in terms of both time and money. Given that economic capabilities differ between countries, those with a high gross domestic product typically have high financial capacities to invest in such risk-reducing technologies[68]. For example, the Netherlands, which adopts a partly protect-open strategy (with storm surge barriers in the Eastern and Western Scheldt), manages the Delta Fund for climate adaptation, with an annual budget of approximately €1.25 billion. Over half of this budget is allocated for investing in new adaptation measures, while the remainder covers measure maintenance[69]. However, countries with a lower economic capacity may be limited to low-capital strategies, highlighting how economic barriers shape feasibility beyond the physical dimension. Similarly, despite the broad physical feasibility of a retreat strategy (Fig. 2), social, political, or economic barriers may limit implementation of the strategy[6,10,16,62]. For a protect-closed strategy,

financial constraints in constructing levees and pumps, or social resistance due to shipping or ecological preferences, may also limit strategy implementation despite its physical feasibility[25]. However, for a strategy such as advance, the opposite may occur, where physical limits of meeting pump capacity requirements and collecting sufficient aggradation material (Fig. 6) may hinder strategy implementation prior to economic or social resistance. Considering the legal dimension of the solution space, conflicting interests, actors and structures in the governance systems responsible for implementing adaptation result in additional decision-making complexities[27].

The physical feasibility and the selection of delta adaptation strategies are shaped by multiple physical constraints. Our first order assessment, which focuses on a limited set of options, provides a global overview of adaptation opportunities and limitations. Additionally, we find that the global threat of SLR can lead to resource and technological limits, highlighting the importance of regional and global coordination.

## Methods

### Overview of approach
We map global deltas' physical solution space (PSS) following three IPCC SSP-RCP scenarios (SSP1-2.6, SSP2-4.5 and SSP5-8.5) by 2100. These methods are accompanied by the Supplementary Information, which includes information on the adaptation strategies assessed (see Supplementary Text 1), the spatial extent of the deltas (see Supplementary Text 2), the physical indicators' equations and data sources per adaptation strategy (see Supplementary Text 3), additional support for the thresholds selected (see Supplementary Text 4), the flood risks for deltas and the differences between SLR scenarios (see Supplementary Text 5) and additional information on sensitivity analyses and model output comparisons to literature (see Supplementary Text 6).

### The global delta dataset
The global delta dataset contains 769 delta locations represented by polygons that define the delta area as a four-point deltaic extent[46,70] (see Supplementary Text 2 for a visualisation of the deltaic extent). These deltas represent 35% of 2174 global deltas[71] but cover approximately 96% of the global delta geomorphic area (815,247 km² of 847,936 km²). According to the 2022 global population, 348 million people were living on these deltas, which represented 4.3% of the population at the time[72,73].

### Creating physical indicators to assess physical feasibility
We assess adaptation to SLR to reduce coastal flood risks whilst accounting for river flooding, but excluding pluvial flooding. We create physical indicators of feasibility, which are quantifiable requirements that represent key measures associated with each adaptation strategy. For example, under a protect-closed strategy, a key measure is a pump to discharge the mean river discharge into the (elevated) sea, combined with retention areas to store excess water during a high river flow. As such, the physical indicators are pump capacity and the availability of water retention areas to store excess discharge that cannot be pumped out. Under an accommodate strategy, built-up areas are raised, so the physical indicator is the height necessary to raise these areas based on the flood depths in the delta. We use a combination of literature and interviews with experts in coastal engineering and flood resilience solutions to determine the types and proportions of indicators[2,9,12,14,17,20,74] (Table 1). Each physical indicator is calculated using a simple geometric model (see Supplementary Text 3 for physical indicator equations). We run the model under three SLR scenarios following the IPCC's SSP-RCP scenarios (SSP1-2.6, SSP2-4.5 and SSP5-8.5), but do not find significant differences between the outputs and thus focus this study on an intermediate GHG emissions scenario (SSP2-4.5) (see Supplementary Text 5 and results section

**Table 1 | The physical indicators, units and equations (Eqn) per adaptation strategy**

| Strategy | Physical indicator | Unit | Eqn |
|---|---|---|---|
| Advance | Total volume of material required to extend the coastline seaward | $m^3$ | 1 |
| | Sufficient river sediment to aggrade a new coastline over 50 years | % | 2 |
| | Accessible offshore depth for sand mining to fill the new coastline | m | - |
| | Technically feasible pump capacity to pump the mean river discharge and 99th percentile river discharge | $m^3.s^{-1}$ | 3 |
| | Sufficient volume capacity of water retention areas (coastal lagoon, permanent waterbodies and wetlands) to store excess river discharge that isn't pumped | $m^3$ | 4 |
| Protect-closed | Total volume of material required to construct levees along the coast | $m^3$ | 5 |
| | Technically feasible pump capacity to pump the mean river discharge and 99th percentile river discharge | $m^3.s^{-1}$ | 3 |
| | Sufficient volume capacity of water retention areas (permanent waterbodies and wetlands) to store excess river discharge that isn't pumped | $m^3$ | 4 |
| Protect-open | Volume of material required to construct levees along the coast and rivers | $m^3$ | 6 |
| | Suitably sized river mouth width for storm surge barrier construction | m | 7 |
| Accommodate | Feasible height to raise homes based on flood depth of urban areas | m | - |
| Retreat | Sufficient land availability for a planned retreat | ratio | 8 |
| Do nothing | Presence of flood risk | = 0 | - |

Model equations and input variables (using existing data) can be found in Supplementary Text 3.

**Table 2 | The three thresholds applied to the physical indicators of each adaptation strategy and the criterion used to determine whether certain strategies are physically feasible or not**

| Strategy | Physical indicator | Threshold 1 | Threshold 2 | Threshold 3 |
|---|---|---|---|---|
| Advance | Sediment retention rate to aggrade new coastline over 50 years | 40%[85–89] | 20% | 80% |
| | Offshore depth for sand mining | <30 m[90–92] | <15 m | <60 m |
| | Pump capacity (based on mean and 99th percentile river discharge) | <1200$m^3.s^{-1}$[93] | <600$m^3.s^{-1}$ | <2400$m^3.s^{-1}$ |
| | Water retention areas to store excess river discharge | Excess river water storage with 1200$m^3.s^{-1}$ pumps | Excess river water storage with 600$m^3.s^{-1}$ pumps | Excess river water storage with 2400$m^3.s^{-1}$ pumps |
| | Feasibility criterion: ((Pump capacity (mean river discharge) AND offshore depth) OR (Pump capacity (mean river discharge) AND river sediment)) AND (Pump capacity (99th percentile river discharge) AND water retention areas) | | | |
| Protect-closed | Pump capacity (based on mean and 99th percentile river discharge) | <1200 $m^3.s^{-1}$[93] | <600 $m^3.s^{-1}$ | <2400 $m^3.s^{-1}$ |
| | Water retention areas to store excess river discharge | Excess river water storage with 1200$m^3.s^{-1}$ pumps | Excess river water storage with 600$m^3.s^{-1}$ pumps | Excess river water storage with 2400$m^3.s^{-1}$ pumps |
| | Feasibility criterion: Pump capacity (mean river discharge) AND (Pump capacity (99th percentile river discharge) AND water retention areas) | | | |
| Protect-open | River mouth width for storm surge barrier construction | <9 km[94] | <4.5 km | <18 km |
| Accommodate | Flood depth of urban area for home raising | <1 m[60,95] | <0.5 m | <2 m |
| Retreat | Land availability for a planned retreat | Non-flooded area within delta[95–97] | Non-flooded urban area within delta | Outside of delta |

Threshold 1 represents the current known threshold (see Supplementary Dataset 1 for database of existing measures in practice and their magnitudes) and threshold 2 and 3 represent low-resource and innovative versions of the current known measures, respectively.

Thresholds are not applied to every physical indicator. For example, the volume of material required is an absolute value and thresholds to assess the feasibility of this measure are only applied to the material source.

titled 'Physical characteristics and adaptation thresholds versus climate change').

## Applying thresholds to the physical indicators

We apply three thresholds to each physical indicator to assess the physical opportunities or constraints of the strategy (Table 2). Threshold 1 is the current known scale of adaptation measures, which is either the maximum known (e.g. the largest existing storm surge barrier) or most commonly used (e.g. home raising through stilts) magnitude of a measure in practice (see Supplementary Dataset 1 for a database of measures in practice and their magnitude). Threshold 2 is a low-resource version of the measure, which can be adopted under limited resource (material) conditions. Threshold 3 is a more advanced version of the current known measures and would require additional innovation, collaboration, or resources to adopt. We assume that such advanced or innovative measures provide more opportunities for delta adaptation; however, what is considered 'innovative' may differ between deltas (see Supplementary Text 4 for rationale on the thresholds).

## Sensitivity analysis and model output comparison

To evaluate the model reliability, we conducted sensitivity analyses by changing the input parameters to values well beyond plausible ranges. This test confirmed that the model responds appropriately to extreme values, with large inputs contracting the PSS and smaller inputs expanding the PSS (see Supplementary Text 6). However, we also test parameter uncertainty by using plausible projected ranges for river and sediment discharge by 2100 from existing literature[50–52]. We

evaluate how projected changes in these input parameters influence the PSS of global deltas and confirm that changes in individual deltas' PSS do not alter the overall global adaptation trends. However, these changes may have implications for local-scale decision-making in 3–8% of deltas (see Supplementary Text 6 for examples of deltas impacted). We further test how higher amounts of future SLR (2 m and 4 m) influence the PSS. We find that SLR beyond the likely range for this century contracts the PSS of many deltas, specifically decreasing the physical feasibility of the advance, accommodate and retreat strategies (see Supplementary Text 5 for further explanation and Supplementary Fig. 5 for a visualisation of this contraction).

The three thresholds applied (Table 2) provide insight into delta adaptation potential, while also testing the model sensitivity. As the threshold changes from low-resource to innovative, the PSS expands. If the innovation threshold is increased from a twofold to a tenfold change, the PSS expands for some deltas; however, the overall outcomes remain largely consistent. In most cases, strategies that are physically unfeasible under all three thresholds remain unfeasible even at higher thresholds (see Supplementary Text 6).

Finally, we compare the model outcomes with existing literature on known and future delta adaptation plans for 10 deltas. We find that the strategies we identify as physically feasible for certain deltas align with the currently implemented and planned future strategies (see Supplementary Text 6 and Supplementary Dataset 2).

### Considerations for future solution space assessments

While this paper informs the PSS of adaptation in global deltas, there are several considerations that should be addressed in future work. Here, we assessed one to five key measures for each adaptation strategy; however, additional measures can be assessed that may expand the PSS. These include the construction of closure dams, converting agriculture to fish farming, emergency flood shelters, planned no-build zones or nature-based solutions (NBS) such as mangrove belts[75–78]. For NBS, the rate of SLR needs to be considered, since NBS can typically only adapt to gradual rates of SLR and may lose effectiveness under higher climate scenarios[79–82].

Future PSS assessments should also assess the physical feasibility of adaptation strategies at a sub-delta scale, since different areas may require different strategies based on flood risks in the delta. Incorporating strategy complementarity, for example, using land creation to facilitate a relocation elsewhere, may provide a more holistic assessment of the adaptation opportunities relevant for policymakers. A smaller-scale assessment may also support higher resolution flood maps, which could reduce potential overestimations of flood extents in urban areas and provide more detail on projected flood risks (see Supplementary Text 3 for more information on flood maps). Future research should also integrate delta morphology analyses with adaptation feasibility assessments for a comprehensive understanding of how deltas will change over time and which strategies will remain physically feasible in the future.

### Reporting summary

Further information on research design is available in the Nature Portfolio Reporting Summary linked to this article.

## Data availability

The input data of the geometric model and the data generated in this study have been deposited in the figshare repository[83].

## Code availability

The Python scripts developed for the data analysis have been deposited in a CodeOcean repository[84] and are supported by the README.txt in the supplementary files.

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

## Acknowledgements

K.G.L. acknowledges support from Utrecht University, Deltares and the Delta Climate Centre (DCC). J.H.N. acknowledges support from NWO VI.Vidi.233.166. We thank Ilse van den Broek for graphically designing the adaptation strategies figure.

## Author contributions

K.G.L., J.H.N and M.H. conceptualised the research question. K.G.L. wrote the original draft of the manuscript. K.G.L. performed the math-ematical calculations for the model and analysed all data. J.H.N., M.H. and G.W. reviewed and edited the manuscript. G.W. provided data on global flood maps. All authors contributed to the manuscript.

## Competing interests

The authors declare no competing interests.
