## [Transparent Peer Review file · Nature Communications]

Physical limits of sea-level rise adaptation in global river deltas

Corresponding Author: Ms Kiara Lasch

Version 0:

Reviewer comments:

Reviewer #1

(Remarks to the Author)

This manuscript evaluates a range of possible solutions for adaptation to future sea-level rise conditions for a large number of river deltas globally. Specifically, the authors base their assessment on a first-order analysis of five different adaptation strategies, showing the number and type of physically feasible adaptation strategies across deltas. The results provide an overview on which deltas could adapt to future conditions using existing resources and technologies, and which require technical innovation or coordination with nearby territories to adapt to future flood risks.

This is the second time that I have to opportunity to review this manuscript, and I am glad to see that the authors have well addressed all my previous comments. The additional analyses have further improved the methodology and the robustness of results. The authors have also improved the description of the modelling steps and clarified the rationale behind some modelling assumptions.

My only remark is that the manuscript seems not to have a data availability section. In case this is not requested at a later stage of the submission process, I would strongly encourage the authors to add this section. Apart from this technical issue, in my opinion the manuscript can be published in its present form

Francesco Dottori
IUSS Pavia, Italy

(Remarks on code availability)

I quickly examined the code and data provided by the authors but I did not try to install and run the code as I do not have time for a complete check.

Input and output data seem to be available.

I could not find README or similar instruction files explaining the structure of the code and input-output data. If this is the case, I would recommend adding these instructions.

Reviewer #2

(Remarks to the Author)

It is the second time I review this paper, following my earlier review of its prior version submitted to Nature Climate Change. This study delivers a valuable global-scale assessment of the physical solution space (PSS) for sea-level rise (SLR) adaptation across 769 river deltas, effectively addressing a critical gap in understanding the feasibility of five core adaptation strategies. The systematic quantification of material requirements, hydrodynamic constraints, and geomorphic thresholds—paired with the integration of global datasets like SWORD river networks and MERIT DEM—establishes a novel comparative framework. Key findings, such as the universality of at least one feasible strategy per delta, material limitations for the “advance” strategy, and the role of delta size and urbanization in constraining PSS, offer actionable insights for prioritizing adaptation resources. Compared with the prior version, I do observe specific improvements, including the revised calculation method for river widths of 369 deltas. However, the major concerns I raised earlier remain unresolved, and new issues have emerged with unexplained updates of parameters and results in the current manuscript, which require further attention.

First, the definitions of the five adaptation strategies still lack refinement to align with real-world implementation. The distinction between “protect-open” and “protect-closed” remains overly narrow, as it focuses primarily on river-mouth structures (storm surge barriers versus dams) while overlooking broader components of practical flood defense systems. For example, “protect-closed” systems like the Dutch Delta Works typically integrate pumps with riverine levees to manage inland flooding, yet the current model excludes riverine levees from the protect-closed strategy—this omission likely underestimates total material requirements for levee construction and overestimates required pump capacity, given that riverine levees reduce the volume of water needing discharge. Conversely, a “true protect-open” strategy, common in practice (e.g., the Mississippi River’s IHNC Surge Barrier system), usually protects both coastlines and inland rivers with dikes while keeping the river mouth open, but the current model fails to explicitly incorporate river dikes into the protect-open strategy, weakening its alignment with real engineering solutions. For the “advance” strategy, the assumption of 100% pumped drainage of river flows (due to fully blocked outlets) is overly restrictive; real-world land reclamation projects often use partial closures or engineered channels to maintain natural river discharge, reducing pump demands to managing pluvial flooding rather than total river flow. This unrealistic assumption artificially limits the feasibility of the advance strategy, and revising these definitions and their associated indicators (such as adjusting pump capacity thresholds for advance to reflect partial drainage) is essential to improve the validity of the PSS assessment.

Second, the assumption of static river discharge (outlined in Suppl. Text SI3 of the current manuscript) remains problematic. Climate change is projected to alter precipitation and runoff patterns. For regionally vulnerable deltas—such as monsoonal deltas like the Ganges-Brahmaputra-Meghna—discharge variability could be much higher, directly impacting pump capacity requirements for protect-closed and advance strategies (higher discharge increases the need for larger pumps) and sediment supply for the advance strategy (altered discharge changes sediment transport and retention rates). The lack of sensitivity analyses incorporating projected discharge variability undermines the robustness of physical feasibility assessments; a stratified sensitivity analysis, for instance by delta climate zones (tropical, temperate, arid) using region-specific discharge projections, would better capture these uncertainties and strengthen the findings.

Third, Figure 2a in the current manuscript suffers from poor visual discriminability, with thin lines and overlapping colors making trends difficult to decipher. A critical discrepancy further emerges between the figure and the text: the retreat strategy (represented by the outermost line) appears to be 100% feasible across all deltas, yet the text notes that 2% of deltas require innovation to implement retreat (e.g., relocations outside the delta). This inconsistency points to flawed graphical representation, and revisions are necessary.

Fourth, several key parameters and results have changed between the prior version and the current manuscript, with no explicit justification provided. Most notably, for the “advance” and “protect-closed” strategies, Threshold 3 (Innovative option) for pump capacity has been reduced from 12,000 m³/s (prior version) to 2,400 m³/s (current version)—an 80% reduction that directly alters the feasibility of these strategies. Yet, no rationale for this adjustment—such as literature citations of maximum operational pump capacities, technical limits of current pumping technology, or expert consensus—is provided in the main text, Supplementary Text SI4 (intended to support threshold selections with a database of existing measures), or associated datasets. Similarly, the percentage of deltas with all five strategies feasible in low-resource form under the SSP2-4.5 scenario has increased from 12% (prior version) to 38% (current version), with no explanation for this large discrepancy (e.g., whether it stems from revised data inputs, adjusted thresholds, or model improvements)—this lack of transparency erodes confidence in the result’s reliability. The authors must explicitly document the rationale for all parameter changes and cross-validate new thresholds with existing engineering practices or literature.

In summary, the study addresses an important and timely topic, but unresolved issues with strategy definitions, static discharge assumptions, and methodological transparency—paired with new concerns about unjustified parameter changes—limit its current impact. Addressing these comments will substantially improve the manuscript’s scientific rigor and alignment with real-world adaptation needs, ensuring its findings are both credible and actionable.

(Remarks on code availability)

Reviewer #3

(Remarks to the Author)

This paper sets out a novel global assessment of the physical solution space (PSS) for adaptation of about 800 river deltas to sea level rise (SLR). It triangulates three sources of evidence: physical characteristics of deltas; SLR projections to 2100 (3 SSP scenarios); and three 'scales of adaptation' (assumptions about technical feasibility of adaptation options). It investigates 5 SLR adaptation strategies, including retreat. The results show that '...nearly every delta has at least one physically feasible adaptation strategy'. That seems to be an important result in an interesting and well-constructed paper. The point that this is a first-stage 'physical/engineering' assessment is clearly made and invites additional studies to understand the nature of financial, nature, political and cultural constraints of the proposed measures. Three main comments: 1. The paper presents results for SSP2-4.5 only, arguing that SLR variance is less important than physical characteristics (depth) of deltas to adaptation option feasibility. It would be good to have this more explicitly discussed in the paper, not just in supplementary material (it is an unexpected result). 2. The 'physical indicator thresholds' relate to soft adaptation limits and it would be useful if the values given in Table 2 could be justified. 3. Figure 2 shows that protect-open (levees and storm-surge barriers) is an option that is available more or less everywhere (97% of deltas). This also seems a surprising result, and it may be useful to provide some idea of the cost and time required to put these options in place. This is clearly not an economic feasibility study, but as the paper acknowledges, the finance is likely to be a constraint

in many cases and an indication of the scale of this would be useful.

(Remarks on code availability)

Version 1:

Reviewer comments:

Reviewer #1

(Remarks to the Author)

No further comments from my side, in my opinion the paper can be published as soon as the other referees agree

(Remarks on code availability)

Reviewer #2

(Remarks to the Author)

This is the third time that I have the opportunity to review the paper, and I am glad to see that my previous comments have been well addressed. My only remaining recommendation is to remove the current Fig. 2b. While I acknowledge that this figure has been improved compared to the previous version, its readability remains insufficient. Moreover, it offers very little new information beyond what is already clearly presented in the well-structured Fig. 2a.

(Remarks on code availability)

Reviewer #3

(Remarks to the Author)

This revision has taken seriously the comments and criticisms made by the three reviewers of the earlier draft and been substantially strengthened. This includes some reanalysis as well as clarification, with new sections added to the main text. I continue to believe this is an innovative paper that makes an important contribution to the debate about adaptation to climate change and SLR. The paper invites a further economic and ecological analysis of the consequences of the adaptation strategies tested in this technical analysis.

(Remarks on code availability)

Rebuttal: Physical limits of sea-level rise adaptation in global river deltas

Kiara G. Lasch, Jaap H. Nienhuis, Gundula Winter, Marjolijn Haasnoot

Reviewers Comments:

Reviewer #1 (Comments for the Author):

This manuscript evaluates a range of possible solutions for adaptation to future sea-level rise conditions for a large number of river deltas globally. Specifically, the authors base their assessment on a first-order analysis of five different adaptation strategies, showing the number and type of physically feasible adaptation strategies across deltas. The results provide an overview on which deltas could adapt to future conditions using existing resources and technologies, and which require technical innovation or coordination with nearby territories to adapt to future flood risks.

This is the second time that I have to opportunity to review this manuscript, and I am glad to see that the authors have well addressed all my previous comments. The additional analyses have further improved the methodology and the robustness of results. The authors have also improved the description of the modelling steps and clarified the rationale behind some modelling assumptions.

My only remark is that the manuscript seems not to have a data availability section. In case this is not requested at a later stage of the submission process, I would strongly encourage the authors to add this section.

We thank Francesco for his positive review and are pleased to see he is satisfied with how we addressed his previous comments. We have now added a data availability section to the manuscript.

Apart from this technical issue, in my opinion the manuscript can be published in its present form.
Francesco Dottori
IUSS Pavia, Italy

(Remarks on code availability):

I quickly examined the code and data provided by the authors but I did not try to install and run the code as I do not have time for a complete check.

Input and output data seem to be available.

I could not find README or similar instruction files explaining the structure of the code and input-output data. If this is the case, O would recommend adding these instructions.

All the data and scripts are available through Code Ocean. We have now added a README.txt file to explain the structure of the code and the IO data.

Reviewer #2 (Comments for the Author):

It is the second time I review this paper, following my earlier review of its prior version submitted to Nature Climate Change. This study delivers a valuable global-scale assessment of the physical solution space (PSS) for sea-level rise (SLR) adaptation across 769 river deltas, effectively addressing a critical gap in understanding the feasibility of five core adaptation strategies. The systematic quantification of material requirements, hydrodynamic constraints, and geomorphic thresholds—paired with the integration of global datasets like SWORD river networks and MERIT DEM—establishes a novel comparative framework. Key findings, such as the universality of at least one feasible strategy per delta, material limitations for the “advance” strategy, and the role of delta size and urbanization in constraining PSS, offer actionable insights for prioritizing adaptation resources. Compared with the prior version, I do observe specific improvements, including the revised calculation method for river widths of 369 deltas.

We thank reviewer 2 for reviewing our paper for the second time. We appreciate their recognition of the paper’s contribution to addressing a gap in the knowledge of the physical feasibility of adaptation globally, particularly highlighting the value of our systematic approach and novel comparative framework. We are pleased that the improvements in the revised version were noted and believe we have now, with our extra analysis, addressed their comments comprehensively.

However, the major concerns I raised earlier remain unresolved, and new issues have emerged with unexplained updates of parameters and results in the current manuscript, which require further attention.

First, the definitions of the five adaptation strategies still lack refinement to align with real-world implementation. The distinction between “protect-open” and “protect-closed” remains overly narrow, as it focuses primarily on river-mouth structures (storm surge barriers versus dams) while overlooking broader components of practical flood defense systems. For example, “protect-closed” systems like the Dutch Delta Works typically integrate pumps with riverine levees to manage inland flooding, yet the current model excludes riverine levees from the protect-closed strategy—this omission likely underestimates total material requirements for levee construction and overestimates required pump capacity, given that riverine levees reduce the volume of water needing discharge.

Regarding the definitions of the five adaptation strategies, we follow the IPCC framework (Glavovic et al. 2022). The separation between protect-open and protect-closed follows (Haasnoot et al. 2020; van Alphen, Haasnoot & Diermanse 2022) and is intentionally implemented uniformly across the whole delta. The protect-closed strategy reflects a fully closed connection between the river and the sea where discharge is pumped out and the entire coastline is protected. We acknowledge that, in practice, hybrid strategies can be implemented but then the solution space is unconstrained, and this does not answer our main research question about the physical limits of any individual strategy.

We acknowledge that excluding riverine levees for “protect-closed” can lead to large pump capacity requirements. However, we determine the pump requirements using the **mean annual river**

discharge per delta to determine whether at least the mean annual discharge can be pumped out nearly every day of the year (much less than extreme discharges such as the annual maximum or a 1-in-100 year discharge). This is a conservative assumption and could underestimate the pump capacity requirements during extreme flows.

To address the reviewer comment, we have now included a global assessment of flood retention capacity in the protect-closed strategy and in the advance strategy. We now evaluate whether deltas have sufficient inland retention capacities to store excess water that cannot be pumped during high-flow periods. This assessment calculates the excess water following a 99th percentile river discharge (typical yearly flood) which is a modelled value from the Water Balance Model reanalysis between 1980 and 2012 (Cohen et al. 2013), and uses spatial data to determine whether this water can be stored in existing lakes or wetlands. This additional analysis makes the assessment of the protect-closed strategy more robust by evaluating both mean river conditions as well as floods. We have now added the results from this analysis to the manuscript, the relevant equations to the Supplementary Text SI3, and the respective physical indicator to the methods section (see below and also highlighted in the manuscript).

Text added to results (Line 123) “While current pumping capacities are sufficient to manage mean river discharge for majority of global deltas (91%), these capacities are exceeded during high river flows, necessitating the storage of excess discharge in designated retention areas or larger pumping capacities. Under a 99th percentile river flow event (Cohen et al. 2013; Nanditha & Mishra 2022), protect-closed is physically feasible for most deltas (75%) as the discharge can either be entirely pumped out based on current pump capacity capabilities, or excess discharge can be stored in retention areas. Alternatively, excess river water can be drained under gravity. However, future SLR may limit gravity drainage, which may increase pump capacity requirements beyond what is current possible and further constrain the physical feasibility of protect-closed in these deltas.”

Text added to Supplementary Text SI3 (Line 151): “The required water retention capacity to store excess river discharge during a flood event (maximum river discharge), assuming a 3-day flood event is calculated as described above (Eqn. 4). However, retention areas in a protect-closed strategy only include permanent water bodies ($A_{ret,waterbodies}$) and wetland areas ($A_{ret,wetlands}$) extracted from the global land use dataset (Buchhorn et al. 2019).”

Conversely, a “true protect-open” strategy, common in practice (e.g., the Mississippi River’s IHNC Surge Barrier system), usually protects both coastlines and inland rivers with dikes while keeping the river mouth open, but the current model fails to explicitly incorporate river dikes into the protect-open strategy, weakening its alignment with real engineering solutions.

We believe there may be a misunderstanding. We explicitly include river dikes and coastal dikes in protect-open. We also assess the feasibility of storm surge barriers that remain open under normal conditions and close during high water levels. We define protect-open in Figure 1 and specify the physical indicators in Table 1, and more information can be found in the Supplementary Text SI1 and in the equation in the Supplementary Text SI3.

For the “advance” strategy, the assumption of 100% pumped drainage of river flows (due to fully blocked outlets) is overly restrictive; real-world land reclamation projects often use partial closures or engineered channels to maintain natural river discharge, reducing pump demands to managing pluvial flooding rather than total river flow. This unrealistic assumption artificially limits the feasibility of the advance strategy, and revising these definitions and their associated indicators (such as adjusting pump capacity thresholds for advance to reflect partial drainage) is essential to improve the validity of the PSS assessment.

We believe this comment arises from a misunderstanding of our calculations. The advance strategy (also following the IPCC definition in Oppenheimer et al. (2019), Bongarts Lebbe et al. (2021) and Glavovic et al. (2022)) is defined as a sea-level rise adaptation strategy where the coastline is extended seaward, which creates new (reclaimed) land that is fully protected from the sea by levees, but

also protects existing inland areas (otherwise it would not be considered a flood-protection strategy). We agree that the advantage of this strategy is that it can provide space for flood water retention areas through a coastal lagoon, as seen in the Netherlands for future adaptation strategies (van Alphen et al. 2022). Since the new coastline forms a closed system with the retention lagoon, river drainage through pumps remains necessary, similar to the protect-closed strategy.

That said, we acknowledge that the main flood protection benefit (increased space for flood retention) of the advance strategy was not incorporated in the original manuscript, restricting its representation of real-world examples of advance. We have now incorporated flood water retention areas adjacent to the advanced coastline, which stores excess water during high river discharge events when pumping alone is insufficient, and thereby increases the number of deltas that can implement the strategy.

We believe this analysis represents this strategy in a comprehensive way and appreciate this reviewer’s comment to bring this to our attention. Text describing this update has been added to the Methods section (additional physical indicator), the respective equation has been added to the Supplementary Text SI3, and corresponding results have been adjusted accordingly.

Text added to Supplementary Text SI3 (Line 113): “The required water retention capacity to store excess river discharge during a flood event (maximum river discharge), assuming a 3-day flood event, is calculated using:

$$V_{river,excess} = \int_0^{T_{flood}} (Q_{r,max} - PC_{mean}) dt$$

$$V_{ret,needed} = d(A_{ret,advance} + A_{ret,waterbodies} + A_{ret,wetlands})$$

Where river discharge during a flood event ($Q_{r,max}$) is the 99th percentile of discharges, as above (Cohen et al. 2013) and where T_{flood} is 3 days. We assess the storage capacity of potential retention areas ($V_{ret,needed}$), by summing the area in the advanced delta ($A_{ret,advance}$), calculated as a product of offshore distance (2000m) and coastline length (c)(Fig. SI3.1), and permanent water bodies ($A_{ret,waterbodies}$) and wetland areas ($A_{ret,wetlands}$) extracted from the global land use dataset (Buchhorn et al. 2019). Each retention area is assumed to have a water storage capacity of $d=1$ m depth.”

Regarding the additional comment on pluvial flooding: Although we recognize this is a risk in many deltas and other landforms, the focus of our study is sea-level rise adaptation while accounting for river discharges. We therefore do not consider adaptation measures to prevent fluvial flooding. We now write this more clearly in the methods section.

Text added to methods (Line 419) “We assess adaptation to sea-level rise (SLR) to reduce coastal flood risks whilst accounting for river flooding, but excluding pluvial flooding.”

Second, the assumption of static river discharge (outlined in Suppl. Text SI3 of the current manuscript) remains problematic. Climate change is projected to alter precipitation and runoff patterns. For regionally vulnerable deltas—such as monsoonal deltas like the Ganges-Brahmaputra-Meghna—discharge variability could be much higher, directly impacting pump capacity requirements for protect-closed and advance strategies (higher discharge increases the need for larger pumps) and sediment supply for the advance strategy (altered discharge changes sediment transport and retention rates). The lack of sensitivity analyses incorporating projected discharge variability undermines the robustness of physical feasibility assessments; a stratified sensitivity analysis, for instance by delta climate zones (tropical, temperate, arid) using region-specific discharge projections, would better capture these uncertainties and strengthen the findings.

During the first round of reviews, we addressed reviewer 1 and 2's concerns about river discharge variability by conducting a sensitivity analysis that includes future changes to river and sediment discharge, as outlined in Supplementary Text SI3 and SI6.

In this analysis, we use projections of future discharge (Van Vliet et al. 2013; Moragoda & Cohen 2020). Specifically, we considered scenarios with river and sediment discharge increasing by up to 65% or decreasing by 23%, to consider the full range of variability from both basin and global scale projections. The results of this analysis showed that the PSS of only a few of the global deltas (27-66 deltas) had expanded or contracted in response to altered river or sediment discharge values. Although this change does not substantially alter the overall global assessment of the physical solution space, it has important implications for individual deltas.

To better reflect this, in the Supplementary Text SI6, we state how many deltas have contracted or expanded PSSs following the projected changes in river and sediment discharge, respectively. Moreover, we have now added, in the main text (results section "physical characteristics and adaptation thresholds versus climate change"), a small discussion on this projected change and regions that are expected to experience increased or decreased discharges.

Text edited and examples added to Supplementary Text SI6 (Line 352): "Under increased river discharge projections, protect-closed becomes physically constrained in 66 deltas that could previously adopt it under static conditions. Increased future river discharges raise pump capacity requirements that exceed current capabilities, and expand the required retention areas for excess discharge storage, thereby constraining the PSS of these deltas. Examples include the Incomati (Mozambique), Wami (Tanzania) and Noatak (Alaska) deltas which are in regions (East Africa and the Arctic, respectively) that are projected to increase according to previous studies (Van Vliet et al. 2013; Moragoda & Cohen 2020). Under these future conditions, innovation, hybrid or alternative adaptation strategies may be required to adapt to future flood risks. Under combined increased river and sediment discharge projections, the number of deltas that can adopt advance decreases by 51 deltas compared to no sediment or river discharge change. Despite higher sediment loads to aggrade the coastlines, increased river discharges constrain the pump capacity capabilities and decrease the overall physical feasibility of this strategy. Examples include the Jiulong Jiang (China) and Labuk (Malaysia) deltas, which are both located in tropical (monsoon) regions where river and sediment discharges are projected to increase (Van Vliet et al. 2013; Moragoda & Cohen 2020).

Under decreased river discharge projections, we find that 40 additional deltas can adopt a protect-closed strategy given lower pump capacity requirements and lower discharges during extreme events which decreases the amount of space required to retain excess river water. For example, in regions where river discharges are projected to decrease, such as Southern Europe and North America (Van Vliet et al. 2013; Moragoda & Cohen 2020), the PSS (for protect-closed) is projected to expand, as found in the Ebro (Spain) and the Altamaha (USA) deltas, respectively. Under both decreased river and sediment discharge projections, the advance strategy becomes physically feasible for an additional 27 deltas, primarily due to reduced pump capacity requirements. In these deltas, the pump capacity plays a larger role in determining physical feasibility of advance than availability of river sediment to aggrade the coast, given that alternative material sources such as offshore sand remain accessible despite decreased sediment discharges. Examples include the Altamaha (USA) and Kikori (Papua New Guinea) deltas, where river and sediment discharges are projected to decrease, consistent with previous studies reporting these trends for North America and Australia, respectively (Van Vliet et al. 2013; Moragoda & Cohen 2020). While projected increases or decreases of these parameters impacts several individual deltas, which has implications for local scale decision-making, the general adaptation trends remain mostly consistent across the global scale."

Text added to manuscript results (Line 286) "Moreover, climate change is expected to alter future river and sediment discharge (Van Vliet et al. 2013; Dunn et al. 2019; Moragoda & Cohen 2020), which may

impact pump capacity and water retention requirements (protect-closed and advance) and sediment availability (advance). Our sensitivity analysis reveals that changes in these parameters, based on literature (Van Vliet et al. 2013; Dunn et al. 2019; Moragoda & Cohen 2020), impact the PSS of several deltas which influences local decision-making. Increased river discharge values typically constrain the PSS where protect-closed and advance become physically unfeasible without innovation, for example in tropical (monsoon) regions (Van Vliet et al. 2013). Conversely, decreased discharge projections typically expand the PSS, making both advance and protect-closed physically feasible with current technological capabilities and resources, for example, in North America (Van Vliet et al. 2013; Moragoda & Cohen 2020) (see Methods and Supplementary Text SI6.1 for results from sensitivity analysis and examples of deltas).”

Third, Figure 2a in the current manuscript suffers from poor visual discriminability, with thin lines and overlapping colors making trends difficult to decipher. A critical discrepancy further emerges between the figure and the text: the retreat strategy (represented by the outermost line) appears to be 100% feasible across all deltas, yet the text notes that 2% of deltas require innovation to implement retreat (e.g., relocations outside the delta). This inconsistency points to flawed graphical representation, and revisions are necessary.

Thank you for your suggestion, we have improved visibility further by grouping deltas with similar physically feasible strategies together. Figure 2 has now been updated. Please see below the revised figure (left) compared to the previous figure (right).

The retreat strategy appears 100% feasible for the darkest (innovation) threshold because it is. In our results, we state that 2% of deltas require innovation for retreat because this strategy is not feasible under low-resource and current known thresholds, due to a lack of space to retreat to. This can be seen towards the bottom of the figure where the darker shade of red is highlighted (innovation is required), but the lighter shades of red are not coloured in and the row transitions straight to green (Protect-open). Some examples of deltas with this PSS have been highlighted in the figure.

Fourth, several key parameters and results have changed between the prior version and the current manuscript, with no explicit justification provided. Most notably, for the “advance” and “protect-closed” strategies, Threshold 3 (Innovative option) for pump capacity has been reduced from 12,000 m³/s (prior version) to 2,400 m³/s (current version)—an 80% reduction that directly alters the feasibility of these strategies. Yet, no rationale for this adjustment—such as literature citations of maximum operational pump capacities, technical limits of current pumping technology, or expert consensus—is provided in the main text, Supplementary Text SI4 (intended to support threshold selections with a database of existing measures), or associated datasets.

During the first round of feedback, we changed the thresholds for low-resource and innovative conditions following a suggestion from reviewer 1. We do include literature citations of maximum operational pump capacities and technical limits of current pumping technology to support the use of the current known thresholds on which the innovative threshold is based (see Supplementary Table SI4.1 and TableSI4.2 which supports the selection of our thresholds).

In response to reviewer 2’s comment, we have now expanded the literature overview with additional sources on innovation over the last 100 years. We included years of construction and socio-economic status of the countries, which gives us support for the pace of innovation (using past extrapolations) as well as low-resource options (using differences per economic status). In this analysis, we found that some technological capabilities, such as material extracted from dredging or building heights have increased by more than 10-fold in the last 100 years. However, other technologies have changed less (storm surge barriers, levee heights), by approximately 2-fold. As such, we believe a 2-fold increase is a realistic pace of innovation to include, albeit conservative for certain measures. We have now expanded on this in the Supplementary Text SI4.

Text added to Supplementary Text SI4 (Line 286): “Other examples of technological developments over longer timescales include innovation in infrastructural capabilities, which has led to an increase of over 10-fold in building height between 1995 and 2008 (Kayvani no date). Similarly, the volume of material extracted from dredging has increased by nearly 10-fold from 1925 to 2015 (Cooper et al. 2018). This reflects the substantial scale of technological innovations in the last century. However, the scales of other technologies, such as levee heights or storm surge barrier widths, have changed less over the last 100 years. As such, assuming a 2-fold increase in technological capabilities by 2100 is a realistic pace of innovation, albeit conservative for certain measures.”

Finally, we still include an assessment of the previous 10-fold increase of thresholds and elaborate on its influence on the PSS in the Supplementary Text SI6.1 by stating: “Finally, we explore the influence of the chosen innovative threshold on the PSS. We increase the threshold by an order of magnitude, as opposed to a twofold increase, and find that substantial innovation in technological capabilities does not necessarily imply more strategies are physically feasible. While certain adaptation measures, such as 10m stilts following an accommodate strategy, or 12,000m³.s⁻¹ pumps following a protect-closed strategy, increase the PSS for some deltas, the PSS of other deltas remain unchanged due to fundamental physical characteristics. This highlights that innovation alone does not provide more adaptation opportunities. Given these findings, we maintain the assumption of a twofold increase in innovation capabilities for our analysis since it is more realistic by 2100, and avoids overestimating adaptation opportunities.”

Similarly, the percentage of deltas with all five strategies feasible in low-resource form under the SSP2-4.5 scenario has increased from 12% (prior version) to 38% (current version), with no explanation for this large discrepancy (e.g., whether it stems from revised data inputs, adjusted thresholds, or model improvements)—this lack of transparency erodes confidence in the result’s reliability. The authors must explicitly document the rationale for all parameter changes and cross-validate new thresholds with existing engineering practices or literature.

We apologize for the confusion. This change in percentage stems from the adjusted low-resource and innovation thresholds as described in the paragraphs above. We collected evidence on the current

known capabilities of adaptation measures which we used as the “current known” thresholds and created a low-resource version based on this.

We have now added an analysis to assess the current known capabilities in countries with different income levels (see Supplementary Text SI4). We find that data on land reclamations (Table SI4.1) reveal that higher income countries typically fill the same amount of space within a shorter time frame than lower-income counterparts. Moreover, the scale and height of home raising efforts is substantially more in higher income countries than lower income counterparts. Here, we assume that some (lower income) countries would have lower resources than other (higher income) countries, and find that the largest known scales of measures are in higher income countries (on which we base our “current known” thresholds). As such, we believe that the 2-fold difference in thresholds between low-resource and current-known thresholds are conservative compared to the previous manuscript, but more accurate and validated against existing practices. We have now added text to support this threshold selection (See Supplementary Text SI4).

Text added to Supplementary Text SI4 (Line 264): “The “low-resource” threshold is defined as half the value of the “current known” threshold (see Table 2 in manuscript). We use the relation between income level and adaptation capacity to support this threshold selection and apply all thresholds globally. We assume that some (lower income) countries would have lower resources than other (higher income) countries, and find that the largest known scales of measures are in higher income countries (on which we base our “current known” thresholds). Specifically, data from Table SI4.1 shows that home raising and land reclamation efforts in lower-income countries are generally smaller (number of homes and height raised) and less frequent (smaller areas of land over prolonged periods of time) than in higher-income countries by approximately a factor of 2. As such, we believe that the 2-fold difference in thresholds between the low-resource and current-known thresholds are conservative, but validated against existing examples.”

In summary, the study addresses an important and timely topic, but unresolved issues with strategy definitions, static discharge assumptions, and methodological transparency—paired with new concerns about unjustified parameter changes—limit its current impact. Addressing these comments will substantially improve the manuscript’s scientific rigor and alignment with real-world adaptation needs, ensuring its findings are both credible and actionable.

Thank you for all suggestions. We find that the additional analysis, evidence for thresholds selected, and explanations in our manuscript has much improved.

Reviewer #3 (Comments for the Author):

This paper sets out a novel global assessment of the physical solution space (PSS) for adaptation of about 800 river deltas to sea level rise (SLR). It triangulates three sources of evidence: physical characteristics of deltas; SLR projections to 2100 (3 SSP scenarios); and three 'scales of adaptation' (assumptions about technical feasibility of adaptation options). It investigates 5 SLR adaptation strategies, including retreat. The results show that ‘...nearly every delta has at least one physically feasible adaptation strategy’. That seems to be an important result in an interesting and well-constructed paper. The point that this is a first-stage 'physical/engineering' assessment is clearly made and invites additional studies to understand the nature of financial, nature, political and cultural constraints of the proposed measures.

We thank reviewer 3 for their positive feedback regarding the structure and main messages of this paper.

Three main comments: 1. The paper presents results for SSP2-4.5 only, arguing that SLR variance is less important than physical characteristics (depth) of deltas to adaptation option feasibility. It would be good to have this more explicitly discussed in the paper, not just in supplementary material (it is an unexpected result).

In the results section of our manuscript, we have two sections that elaborate on this. The first section discusses how the adaptation thresholds play a stronger role in determining physical feasibility than SLR. The second section is titled “physical characteristics versus climate change” and discusses how the PSS is more sensitive to delta’s physical characteristics (for example, depth) than the projected SLR under different scenarios. We have now merged these paragraphs into a section titled “Physical characteristics and adaptation thresholds versus climate change” to avoid confusion. In this section, we have now elaborated on the sensitivity of the PSS constraints based on the delta characteristics and measure thresholds (the scale of adaptation) compared to climate change (SLR).

Text added to results section (Line 251): “We find that higher sea levels increase flood risks and shrink the PSS, in agreement with previous studies (Oppenheimer et al. 2019; Haasnoot, Lawrence & Magnan 2021; IPCC 2022). However, this change is relatively small compared to our modelled differences in physical feasibility based on the scale of adaptation measures (i.e. thresholds). We find that the scale of the adaptation measures has a greater influence on the physical feasibility of adaptation strategies than the SLR projections for 2100 (see Supplementary Text SI5 and Fig SI5.1 for a comparison between climate scenarios and thresholds). For instance, the scale of raising infrastructure following accommodate, namely by 0.5m, 1m or 2m, exceeds the median projected global mean sea-level rise for 2100 under SSP1-2.6, SSP2-4.5 and SSP5-8.5 scenarios (0.4m, 0.47m, 0.63m, respectively)(Church et al. 2013). Moreover, the feasibility of offshore dredging to collect sand at depths of 15m, 30m, or 60m is more constrained by technological capabilities to reach the sand than by a 0.5m sea-level rise.

Our assessment further indicates that the current physical characteristics of the deltas (e.g. size, elevation, river discharge as used in this study) are a stronger determinant of the physical feasibility of strategies by 2100 than the projected SLR. For example, the physical feasibility of flood risk protection measures, such as storm surge barriers, depends more on the river width which varies from narrow (100m) to wide (10,000 m) across deltas, than on SLR following future climate scenarios by 2100. Thus, in deltas with wide rivers, protect-open is more physically constrained by the delta characteristics than SLR. Building on this understanding ...[rest of paragraph].

Under higher amounts of SLR beyond the likely range for this century or after 2100, SLR is expected to have a larger influence on the PSS (see Supplementary Text SI5 and Fig SI5.2 for sensitivity analysis with higher SLR). Moreover, climate change is expected to alter future river and sediment discharge (Van Vliet et al. 2013; Dunn et al. 2019; Moragoda & Cohen 2020), which may impact pump capacity and water retention requirements (protect-closed and advance) and sediment availability (advance). ... [rest of paragraph].”

We have now performed an additional sensitivity analysis to support the claim above that larger SLR will influence the PSS. We test how even greater SLR values (2m and 4m) impact the PSS of global deltas (see methods and findings in Supplementary Text SI5 and Fig. SI5.2).

Text added to methods section (Line 460) “We further test how higher amounts of future SLR (2m and 4m) influences the PSS. We find that SLR beyond the likely range for this century contracts the PSS of many deltas, specifically decreasing the physical feasibility of the advance, accommodate and retreat strategies (see Supplementary Text SI5 for further explanation and visualisation of this contraction).”

Text added to Supplementary Text SI5 (Line 323): “In the future, high-end estimates of SLR find that sea-levels may rise to between 0.9m (by 2100) and 2.5m (by 2300) following an SSP1-2.6 scenario or between 1.6m (by 2100) and 10.4m (by 2300) following an SSP5-8.5 scenario (van de Wal et al. 2022). Our sensitivity analysis reveals that 2m and 4m of SLR constrain the PSS for many deltas, specifically for the advance, retreat and accommodate strategies (Fig. SI5.2). Higher sea levels means larger offshore depths for dredging and land aggrading (advance), less available non-flooded land (retreat), and greater

infrastructure elevation requirements following larger flood depths (accommodate) which constrains the physical feasibility of implementing these strategies.”

2. The 'physical indicator thresholds' relate to soft adaptation limits and it would be useful if the values given in Table 2 could be justified.

We agree that it is useful to highlight the sources in the manuscript instead of only in the supplementary text. Thank you for the suggestion. The thresholds have now been justified with a “source” column in Table 2 in the methods section of the manuscript.

3. Figure 2 shows that protect-open (levees and storm-surge barriers) is an option that is available more or less everywhere (97% of deltas). This also seems a surprising result, and it may be useful to provide some idea of the cost and time required to put these options in place. This is clearly not an economic feasibility study, but as the paper acknowledges, the finance is likely to be a constraint in many cases and an indication of the scale of this would be useful.

Indeed, while protect-open may be physically feasible, it is a costly strategy. We have now added more on this to the discussion section.

Text added to discussion (Line 370): “For example, while protect-open is physically feasible across majority of deltas to protect against coastal flooding (Fig. 2), it is a costly strategy in terms of both time and money. Given that economic capabilities differ between countries, those with a high gross domestic product (GDP) typically have high financial capacities to invest in risk-reducing technologies (Tessler et al. 2015) For example, the Netherlands, which adopts a partly protect-open strategy (with storm surge barriers in the Eastern and Western Scheldt) manages the Delta Fund for climate adaptation, with an annual budget of approximately €1.25 billion. Over half of this budget is allocated for investing in new adaptation measures, while the remainder covers measure maintenance (Deltacommissaris 2021). However, countries with a lower economic capacity may be limited to low-capital strategies, highlighting how economic barriers shape feasibility beyond the physical dimension.”

Reference list

- Alphen, J. van, Haasnoot, M. & Diermanse, F., 2022, ‘Uncertain Accelerated Sea-Level Rise, Potential Consequences, and Adaptive Strategies in The Netherlands’, *Water (Switzerland)*, 14(10).
- Bongarts Lebbe, T., Rey-Valette, H., Chaumillon, É., Camus, G., Almar, R., Cazenave, A., Claudet, J., Rocle, N., Meur-Férec, C., Viard, F., Mercier, D., Dupuy, C., Ménard, F., Rossel, B.A., Mullineaux, L., Sicre, M.A., Zivian, A., Gaill, F. & Euzen, A., 2021, ‘Designing Coastal Adaptation Strategies to Tackle Sea Level Rise’, *Frontiers in Marine Science*, 8, 740602.
- Buchhorn, M., Smets, B., Bertels, L., Roo, B. De, Lesiv, M., Tsendbazar, N.-E., Herold, M. & Fritz, S., 2019, ‘Copernicus Global Land Service: Land Cover 100m: collection 3: epoch 2018: Globe’.
- Church, J.A., Clark, P.U., Cazenave, A., Gregory, J.M., Jevrejeva, S., Levermann, A., Merrifield, M.A., Milne, G.A., Nerem, R.S., Nunn, P.D., Payne, A.J., Pfeffer, W.T., Stammer, D. & Unnikrishnan, A.S., 2013, ‘Sea Level Change’, in T.F. Stocker, D. Qin, G.-K. Plattner, M. Tignor, S.K. Allen, J. Boschung, A. Nauels, Y. Xia, V. Bex & P.M. Midgley (eds.), *Climate Change 2013: The Physical Science Basis. Contribution of Working Group I to the Fifth Assessment Report of the Intergovernmental Panel on Climate Change*, Cambridge University Press.

- Cohen, S., Kettner, A.J., Syvitski, J.P.M. & Fekete, B.M., 2013, 'WBMsed, a distributed global-scale riverine sediment flux model: Model description and validation', *Computers and Geosciences*, 53, 80–93.
- Cooper, A.H., Brown, T.J., Price, S.J., Ford, J.R. & Waters, C.N., 2018, 'Humans are the most significant global geomorphological driving force of the 21st century', *Anthropocene Review*, 5(3), 222–229.
- Deltacommissaris, 2021, *The 2022 Delta Programme, Every New Development Climate-Proof; Ministry of Infrastructure and Water Management*.
- Dunn, F.E., Darby, S.E., Nicholls, R.J., Cohen, S., Zarfl, C. & Fekete, B.M., 2019, 'Projections of declining fluvial sediment delivery to major deltas worldwide in response to climate change and anthropogenic stress', *Environmental Research Letters*, 14(8), 084034.
- Glavovic, B.C., Dawson, R., Chow, W., Garschagen, M., Haasnoot, M., Singh, C. & Thomas, A., 2022, 'Cross-Chapter Paper 2: Cities and Settlements by the Sea', in H.-O. Pörtner, D.C. Roberts, M. Tignor, E.S. Poloczanska, K. Mintenbeck, A. Alegría, M. Craig, S. Langsdorf, S. Löschke, V. Möller, A. Okem & B. Rama (eds.), *Climate Change 2022: Impacts, Adaptation and Vulnerability. Contribution of Working Group II to the Sixth Assessment Report of the Intergovernmental Panel on Climate Change*, pp. 2163–2194, Cambridge University Press, Cambridge, UK and New York, NY, USA.
- Haasnoot, M., Kwadijk, J., Alphen, J. Van, Bars, D. Le, Hurk, B. Van Den, Diermanse, F., Spek, A. Van Der, Oude Essink, G., Delsman, J. & Mens, M., 2020, 'Adaptation to uncertain sea-level rise; how uncertainty in Antarctic mass-loss impacts the coastal adaptation strategy of the Netherlands', *Environmental Research Letters*, 15(3).
- Haasnoot, M., Lawrence, J. & Magnan, A.K., 2021, 'Pathways to coastal retreat: The shrinking solution space for adaptation calls for long-term dynamic planning starting now', *Science*, 372(6548), 1287–1290.
- IPCC, 2022, 'Climate Change 2022: Impacts, Adaptation, and Vulnerability. Contribution of Working Group II to the Sixth Assessment Report of the Intergovernmental Panel on Climate Change', in H.O. Pörtner, D.C. Roberts, M. Tignor, E.S. Poloczanska, K. Mintenbeck, A. Alegría, M. Craig, S. Langsdorf, S. Löschke, V. Möller, A. Okem & B. Rama (eds.), Cambridge, UK and New York, NY, USA.
- Kayvani, K., no date, *The evolution of tall buildings: past and present trends*.
- Moragoda, N. & Cohen, S., 2020, 'Climate-induced trends in global riverine water discharge and suspended sediment dynamics in the 21st century', *Global and Planetary Change*, 191, 103199.
- Nanditha, J.S. & Mishra, V., 2022, 'Multiday Precipitation Is a Prominent Driver of Floods in Indian River Basins', *Water Resources Research*, 58(7), e2022WR032723.
- Oppenheimer, M., Glavovic, B.C., Hinkel, J., Wal, R. van de, Magnan, A.K., Abd-Elgawad, A., Cai, R., Cifuentes-Jara, M., DeConto, R.M., Ghosh, T., Hay, J., Isla, F., Marzeion, B., Meyssignac, B. & Sebesvari, Z., 2019, 'Sea Level Rise and Implications for Low-Lying Islands, Coasts and Communities', in H.O. Pörtner, D.C. Roberts, V. Masson-Delmotte, P. Zhai, M. Tignor, E.

- Poloczanska, K. Mintenbeck, A. Alegría, M. Nicolai, A. Okem, J. Petzold, B. Rama & N.M. Weyer (eds.), *IPCC Special Report on the Ocean and Cryosphere in a Changing Climate*, pp. 321–445, Intergovernmental Panel on Climate Change, Geneva, Switzerland.
- Tessler, Z.D., Vörösmarty, C.J., Grossberg, M., Gladkova, I., Aizenman, H., Syvitski, J.P.M. & Foufoula-Georgiou, E., 2015, 'Profiling risk and sustainability in coastal deltas of the world', *Science*, 349(6248), 638–643.
- Vliet, M.T.H. Van, Franssen, W.H.P., Yearsley, J.R., Ludwig, F., Haddeland, I., Lettenmaier, D.P. & Kabat, P., 2013, 'Global river discharge and water temperature under climate change', *Global Environmental Change*, 23(2), 450–464.
- Wal, R.S.W. van de, Nicholls, R.J., Behar, D., McInnes, K., Stammer, D., Lowe, J.A., Church, J.A., DeConto, R., Fettweis, X., Goelzer, H., Haasnoot, M., Haigh, I.D., Hinkel, J., Horton, B.P., James, T.S., Jenkins, A., LeCozannet, G., Levermann, A., Lipscomb, W.H., Marzeion, B., Pattyn, F., Payne, A.J., Pfeffer, W.T., Price, S.F., Seroussi, H., Sun, S., Veatch, W. & White, K., 2022, 'A High-End Estimate of Sea Level Rise for Practitioners', *Earth's Future*, 10(11), e2022EF002751.

Rebuttal: Physical limits of sea-level rise adaptation in global river deltas

Kiara G. Lasch, Jaap H. Nienhuis, Gundula Winter, Marjolijn Haasnoot

REVIEWERS' COMMENTS

Reviewer #1 (Remarks to the Author):

No further comments from my side, in my opinion the paper can be published as soon as the other referees agree.

We thank Reviewer 1 for their approval of our manuscript.

Reviewer #2 (Remarks to the Author):

This is the third time that I have the opportunity to review the paper, and I am glad to see that my previous comments have been well addressed. My only remaining recommendation is to remove the current Fig. 2b. While I acknowledge that this figure has been improved compared to the previous version, its readability remains insufficient. Moreover, it offers very little new information beyond what is already clearly presented in the well-structured Fig. 2a.

We are pleased to see that Reviewer 2 believes their comments have been well addressed, and we thank them for their time and feedback.

Regarding Fig. 2b, we believe that this stacked histogram compliments Fig. 2a by showing the full distribution of physical feasible strategies across deltas. This demonstrates the spread of feasible strategies, showing similarities in the types and number of strategies across deltas which is not distinguishable in Fig. 2a. As such, we opt to keep this sub-figure in our manuscript.

Reviewer #3 (Remarks to the Author):

This revision has taken seriously the comments and criticisms made by the three reviewers of the earlier draft and been substantially strengthened. This includes some reanalysis as well as clarification, with new sections added to the main text.

I continue to believe this is an innovative paper that makes an important contribution to the debate about adaptation to climate change and SLR. The paper invites a further economic and ecological analysis of the consequences of the adaptation strategies tested in this technical analysis.

We are pleased to see Reviewer 3 is satisfied with the changes made. We thank them for their valuable suggestions.